# The Impact of a Virtual Environment for Intergenerational Learning

**Greg Cronan [1], Janna Anneke Fitzgerald [2] , Katrina Radford [3] and Gabriela Di Perna [2,*]**

1    Intergenerational Learning Australia, Sydney, NSW 2059, Australia
2    Department of Business, Strategy and Innovation, Griffith University, Southport, QLD 4222, Australia
3    Department of Employment Relations and Human Resources, Griffith University,
     Nathan, QLD 4111, Australia
*    Correspondence: gabriela.diperna@griffithuni.edu.au

**Abstract:** Few intergenerational programs have focused on virtual learning. This paper explores the impact that a virtual intergenerational learning program had on primary school students and older adults at a residential aged care home. This study reports on the findings from a single case study design involving interactions across two sites, consisting of primary school students (n = 41), teachers (n = 2), a principal (n = 1), plus elders (n = 9) and staff (n = 4) from an aged care home. Questionnaires were completed by all participants, except for the school principal. Observations of the program interactions were captured via video ethnography. Data were also evaluated from semi-structured interviews of students (n = 16), parents (n = 2), teachers (n = 2), a principal (n = 1), elders (n = 9) and staff (n = 5) from the aged care home. The findings reveal that intergenerational programs must pay close consideration to the planned activities, participant characteristics, learning environment, equipment, and facilitator interactions and skills, particularly in a virtual space. In addition, this study finds continued evidence for the success of virtual intergenerational practice programs to reduce social isolation and exclusion, especially when we practice social distancing.

**Keywords:** intergenerational; social inclusion; aged care; primary school; video conferencing

## 1. Introduction

COVID-19 has produced unprecedented demand and need for virtual intergenerational programs to reduce social isolation and improve the communication and educational outcomes of the younger generation, while also improving the social connections and relationships between both generations. Intergenerational programs are defined as activities that foster cooperation, interaction, and exchange between two or more generations (Kaplan and Sánchez 2014). These programs have increased substantially over the past 20 years to include a wider range of generations, an improved evidenced-based curriculum and more hybrid running programs to successfully connect generations (Belgrave and Keown 2018; Canedo-Garcia et al. 2017).

Intergenerational programs have been found to complement lifelong learning intentions between generations (Thompson and Weaver 2016), improve children's behavior towards older adults and those with dementia (Hayes 2003; Heyman et al. 2011), and empower the formation of positive interpersonal relationships between generations (Femia et al. 2008; Hayes 2003; Radford et al. 2016). These programs also address the World Health Organization's (2016) (WHO) healthy ageing agenda through its Active Ageing framework by empowering the creation of age-friendly cities. Consequently, the continuation of such programs during unprecedented times such as COVID-19 is critical, albeit in a virtual nature. This study adds to this growing body of literature by exploring the perceived impact of a virtual intergenerational program between primary school-aged children (aged: 11–12 years) and residents living in a long-term aged care home (commonly referred to as a residential aged care facility). It seeks to answer three research questions:

1. What are the necessary 'ingredients' for a successful virtual intergenerational program?"
2. Can both generations use, and benefit from, virtual intergenerational programs to improve social inclusion and reduce social isolation?"
3. What is the impact of intergenerational programs on participants that are facilitated and hosted from different remote locations using a digital visual communication technology such as video conferencing?"

Accordingly, this paper seeks to extend the knowledge base in implementation science and social inclusion in relation to intergenerational programs.

## 2. The Case for Intergenerational Programs in Australia

Australia's population is ageing at an unprecedented rate. In 2017/2018, there was a 49% increase in the number of people admitted to aged care services in Australia compared to the previous decade (Australian Government Productivity Commission 2019) and the availability of aged care services to the public has more than tripled since its inception. However, the current offering of services within the aged care sector does not necessarily meet the needs of the older population (Vecchio et al. 2018; Australian Government Productivity Commission 2019). This has prompted calls for additional, more innovative services such as intergenerational programs, to become available within the aged care sector. This need was also highlighted in the interim report released by the Australian Royal Commission into Aged Care and Quality, to address the broken and systemic issues of aged care which currently, "fails to meet the needs of our older, often very vulnerable, citizens" (Commonwealth of Australia 2019, p. 1). Indeed, the commissioner reported that "Australia has drifted into an ageist mindset that undervalues older adults and limits their possibilities" (Commonwealth of Australia 2019, p. 1).

Additionally, most older adults in Australia experience at least one of the risk factors for social isolation, which include the migration of their children, death or disability among their social network members, a decline in their physical and cognitive abilities, altered living arrangements, and/or a lack of participation in social and religious activities. Other research has found that approximately one-third to one-half of people aged over 60 in Australia are at risk of experiencing loneliness (Aged & Community Services Australia 2015; Landeiro et al. 2017). A recently published systematic literature review on loneliness in older adults who live independently (Di Perna et al. 2023) found that the main causes of feeling socially isolated or lonely in general include personal factors such as "advancing age, lower educational levels, living alone, being unemployed, poor self-rated health, being widowed/single, having limited physical activity and being female" as well as environmental factors such as meaningful connections and social activity, connection loss as well as loss in autonomy, technology use, depression, pain and anxiety (Di Perna et al. 2023, p. 7). Furthermore, it is important to note, that the list of antecedents may differ for the objective state of social isolation and the subjective experience of loneliness in older adults. While being socially isolated and lonely are two distinct concepts, both can be addressed through an intergenerational program intervention if successfully planned and implemented. This study specifically examines how we can address social isolation through a virtual intergenerational learning program using video conferencing to connect two diverse age groups: older adults residing in a residential aged care home and school-aged children.

## 3. Addressing Social Isolation through Implementing Virtual Intergenerational Programs

Social isolation is a state of complete, or near-complete, lack of contact between an individual and society (Boyd et al. 2018). The impact that social isolation has on one's health is compelling, as research has found both older and younger populations with more diversified social networks have higher overall physical and psychological wellness scores and perceived health status than those with smaller social networks (Beam and Kim 2020;

Hayashi et al. 2020). Reduced social networks and connections were the most significant antecedent reported in a study by Di Perna et al. (2023) amongst those older people living in a community who experienced loneliness. Social alienation and disengagement from social activity were found to be among the highest risk factors for loneliness experienced (Lai et al. 2020; Wong et al. 2017).

Thus, social inclusion programs that allow generations to connect with their same and opposite peer groups such as intergenerational programs, should be considered essential during, and after, the global pandemic. This would enable different generations to reconnect and provide purposeful and meaningful social interactions that may not otherwise have been available.

Social inclusion refers to the process of improving the terms on which individuals and groups participate in society. For both generations, this may require involving interventions such as intergenerational programs that are designed to connect with family, friends, personal interests, and their local community (Commonwealth of Australia 2012). These programs offer participants a sense of belonging and well-being, address prejudicial attitudes and experiences of discrimination, consider the amount and quality of contact with people, and assume participants are willing to volunteer in inclusion activities and that people are willing to advocate for social inclusion. These are all central tenets of social inclusion (Faulkner et al. 2019).

A contemporary and central element of addressing social isolation today through social inclusion programs is ensuring digital inclusion is embedded within the design of these programs. Digital inclusion is defined as having access to and being proficient in the use of digital-based technologies, including but not limited to laptops, the internet, 'smart' tablets, mobile phones, video conferencing, 'apps' and social media etc. (Wilson et al. 2019). In general, rural and regional Australians, the older population as well as Australians with low levels of income, employment and education have been found to be less digitally included than their compatriots (Wilson et al. 2019). While this may present an interesting social challenge on how we "digitally" include these populations within Australia, a simple solution may be to integrate intergenerational programs through already-existing school-based and residential aged care assets, which allow digital inclusion to occur by purposely including these vulnerable groups in these programs. However, the impact and implications of this have not been well examined, and as such this study specifically sought out these vulnerable groups in the initial design of the virtual intergenerational learning program.

When properly planned and implemented, intergenerational learning programs provide mutually beneficial outcomes for both populations involved. For the younger population, these programs have been found to improve attitudes and behaviors towards older adults (Hayes 2003; Heyman et al. 2011), improve empathy levels in children towards older adults (Femia et al. 2008) and improve self-regulation abilities in children (Femia et al. 2008). For the older population, these programs have been found to improve the overall quality of life reported (Jarrott and Bruno 2003; George and Singer 2011) by providing an atmosphere of revitalized interest and social enrichment (Foster 1997) through the social interactions that are present during the sessions, which positively influence their overall psychological and physical well-being (Canedo-Garcia et al. 2017).

When intergenerational programs are properly planned with a strong pedagogy, younger participants enhance their academic, interpersonal communication and social skills and older participants learn about the school curricula, how to communicate with children and youth, and develop skills to support students with their learning and social development (Newman and Hatton-Yeo 2008). In addition, having relationships with older adults in childhood is associated with improved language development, cognitive processes, socioemotional development, and problem-solving skills, which has a lasting impact on health, future learning and life success (McCain and Mustard 1999; Hambrick et al. 2019; Shanker 2012, 2013). Thus, building the social capital surrounding the populations in that area and bolstering social inclusion in the meantime.

Facilitating a successful virtual intergenerational learning program means that consideration is required to integrate the specific school curriculum topics with the abilities of the participants, the environment for the interactions, equipment (for the video conference) and facilitator interaction. Yet, no research explores what the elements of a successful program look like when considering them in different environments and using technology to connect the two groups via a 'virtual space' such as a video conference.

The aim of this research was to propose characteristics and conditions that might enable the process of fostering age-friendly 'virtual' communities by bringing together school children (in this case between the ages of 11 and 12) and older adults residing in residential aged care homes using video conferencing. The outcomes of this research inform policy by providing further evidence that social interventions such as intergenerational practice, even when virtual in nature, assist with improving social isolation in the community, The results thus provide further evidence on the benefits of including service learning into curriculums across Australian schools to provide one avenue for connecting school-age children with older adults in the community or residential aged care homes.

## 4. Methods

### 4.1. Design

This study uses a single case study design which broadly follows Yin's single case study approach (Yin 2011) with immersive fieldwork that enables researchers to gather further insight into how the program works, what the impacts of the program were on participants and how successful the implementation of the intervention was for all involved (Neuman 2006; Swanborn 2010; Yin 2011). This allows a deeper insight for researchers studying the process and outcome factors involved (Denzin and Lincoln 2000; Yin 2008). Accordingly, this section will detail the context and setting of the program before discussing the data collection and data analysis methods used.

### 4.2. Context and Setting

This study was conducted over two school terms in 2019 (22nd July to 20 December) as part of an honors dissertation project and had full ethical clearance (GU 2019/683) from the university (Cronan 2019). The program ran 45 min a week, typically at the same time and on the same day and involved students from a Year 6 class in Term 3 (22 July to 27 September) and a different Year 6 class of students in Term 4 (14 October to 20 December), attending a school in the Western suburbs of Sydney, Australia. The number of students varied (due to other commitments, illness, etc.) from 4–6 males and 4–6 females (n = 12) each week in Term 3, 2019. The mean age of these students was 11 years and 6 months (Range 10.5–12.5 years). During Term 4, 12–14 males and 12–15 females (n = 29) aged 10–12 years attended each of the video calls. At the residential aged care home 3–4 males and 4–5 females (n = 9) aged between 71 and 95 participated in the video calls during school Term 3 and Term 4. The mean age was 85 years and 4 months.

The school was chosen due to the staff and leadership team having a reputation for being innovative. Students and staff from the school had regularly visited the residential aged care home for various events. The senior adults and staff at the residential aged care home were informed about the program and enthusiastically chose to participate.

Permission for student participation in the video conferences and recordings was requested in writing from their parents or legal guardians. The students were also asked to provide their consent to participate. The older adults (or their legal guardians) and staff at the residential aged care home also completed a participant consent form.

In addition to the technical support provided by the first named author of this paper at the time of the intervention, there were two adult facilitators involved: the teacher of the Year 6 students and the facilitator (Wellness Coordinator) at the residential aged care facility.



*4.3. Equipment Involved*

At the aged care facility, a video conference hardware codec, motorized camera (with remote control of panning, tilt and optical magnification operated by the aged care facilitator), touchscreen control panel, 4G wireless modem and 65″ TV was set up in a coffee room area. Four wireless speakerphones (an electronic device with an integrated microphone and loudspeaker) were placed on tables amongst the older adults and connected wirelessly to the laptop.

In the school classroom, a laptop with an internal camera was placed on a mobile cart and connected to a 65″ TV plus a 4G wireless modem. Four wireless speakerphones were placed on tables amongst the students and connected to the laptop. All sessions were recorded using cloud-based video conferencing tools for analysis at a later time. A test video call was conducted between the school classroom and residential aged care home with the older adults, teacher and first author of this paper to ensure speech was clear, intelligible, and loud enough for the older adults, especially those with diminishing hearing. The Wellness Coordinator at the residential aged care home was also trained on how to use the remote control device for the motorized camera. Confirming the video and sound from the classroom and residential aged care home were of the highest standard possible was important to ensure the first video call with the students and older adults started as a good experience.

*4.4. The Intergenerational Program*

The recruitment process for teachers and students commenced in Term 2 with an overview of the intergenerational learning program delivered to the principal and school leadership team. Upon agreement to continue, the school leadership team invited two Year 6 teachers to attend a presentation about the program. A second presentation about this program was then provided to students in their classroom, in the presence of their teacher. This included information about the format for each video call. This process was repeated for students and teachers in Term 4. At the end of the presentation, the teacher and students discussed this program. An information letter was provided to every student along with a participant consent form. For those students, who were not comfortable being involved in the activity, a different learning activity was provided. In the group of students for Term 3, only one parent did not provide consent. The parent stated, He "*can work on more relevant schoolwork. We'd like him to use his school time more productively*". This student was provided with a learning activity in another room during the video calls. All parents and students consented to participate in Term 4.

The recruitment process for older adults commenced at the same time with the Facility Manager and Wellness Coordinator at the residential aged care facility. The staff and residents at this facility were recruited due to pre-existing interactions with the school in addition to recommendations from the principal and the school's leadership team. However, prior to the commencement of this program, the residents had not met the students who were to be involved in this program. After approval to participate was provided by senior managers of the aged care organization, an overview of this program was presented to the residents (aged 71–95 years). The older participants were screened and recruited by the Wellness Coordinator and Facility Manager due to their knowledge of these residents' personalities and health conditions. They had known some of the nominated residents for years and others only for a few weeks. In addition, their communication abilities and behavior were deemed by these staff members to be suitable for interactions with students online. Older adults were included in the study if they had reasonable hearing (subjectively described and confirmed by staff), average eyesight, and good cognitive and speech skills.

The program was facilitated for 45 min at the same time and typically on the same day of the week, five times per school term resulting in an overall total of 9 video calls across two school terms. In school Term 3, the discussions focused on healthy living as part of the personal development health and physical education (PDHPE) curriculum taught during that term. The students asked the older adults a range of questions about the types

of food available, what was eaten, types of sports and recreational activities, lifestyle, and cost of living when the older adults were in the latter years of primary and early years of high school. Responses by the older adults were handwritten by the students into their notebooks and 'digital tablet devices' for reference when completing their assignments.

In school Term 4, another cohort of students commenced with the curriculum topic being English. Students were required to write poems and limericks after interacting with the older adults. The task for students was to write a limerick with 3 verses, each with 4 lines. An optional extended task was to write 5–6 verses for students who chose this activity. These assessments needed to include comments by the older adults about the discussion topics and details from the stories they told about their experiences of these periods in history. For students who struggled with this task, an easier activity of writing an acrostic poem with a total of 5–6 lines was provided.

During each video call, four to five sets of two students took their turn as hosts, in front of the camera. A few days before each week's video call, the teacher would ask all students if they were interested in the hosting role like TV presenters. Some volunteered, while others accepted her invitation. Their role was to ask questions and respond to comments from the older people. The other students were seated behind the two 'host students' in groups of three tables and were viewable via the web camera in the laptop. Throughout each video call, each set of two students 'hosted' the video call for 7–10 min. They would then say goodbye and introduce the next two 'host students.'

Each of the groups of students behind the two main 'host' students also interacted with the older adults during each video call by asking questions prepared by students or responding to comments from the older adults. A few days before each session, the older participants at the aged care home were provided with a list of questions prepared by the students that were going to be asked in the subsequent video call.

To maintain the attentiveness and active participation of all students throughout the duration of the video call, they were asked to write the following data into four lists: (1) new words, (2) new phrases and (3) vocalizations spoken by the older adults such as giggles, laughter, 'mmm' and 'aha' along with (4) observations of non-verbal communication from the older adults like smiles, hand, and facial gestures etc. The facilitator ensured each older person could view the TV, could hear the students and could be 'seen' by the motorized video conference camera. The Wellness Coordinator using the motorized web camera at the residential aged care home also zoomed in on the older person as they were speaking.

At the end of the video call each week, a questionnaire was completed in writing by the students. These data captured information on the quality of the video, sound, and speech, highlights of the video call and suggestions to improve the next interaction. The questionnaire process was supervised by the teacher and first author of this paper. These data were then entered by the students into an online form.

*4.5. Participant Characteristics*

Due to the availability of the students and interest from students, parents, and legal guardians, this 'convenience' sample consisted of children aged between 10.5–12.5 years. Their cultural backgrounds were from Australia, Malta, Lebanon/Middle East, Asia, India and one student of Aboriginal heritage. These participants ranged in academic abilities. A few of the students had behavioral and cognitive challenges. While most students lived with at least one parent, there was one student who lived at either his grandmother's home or at the home of his aunt. In addition, all but one student had some contact with their grandparents.

The older participants ranged from 71–95 years and were included in the study if they had reasonable hearing (subjectively described and confirmed by staff), average eyesight, and good cognitive and speech skills. Three of the older participants identified as Australian. Other participants had family heritages from Russia, Greece, Mauritius, Germany, Malta and England. All older adults had some form of hearing condition, one had diminished eyesight, and another had lost her eyesight. All could speak English

reasonably well. Those who had English as a second language were less confident to speak. Some older adults preferred not to speak but enjoyed either listening to and/or seeing the students on the TV. In addition, one senior adult had early stages of dementia, one had Parkinson's disease and another one had multiple sclerosis and required a wheelchair. Further, the remaining three older participants had varying levels of pain associated with limited walking and mobility.

### 4.6. Data Collection

Data were collected in written form (paper-based and online) and via video recordings. Prior to all participants commencing this program, they were informed that their participation was voluntary, that the questionnaires were not a test, that there were no wrong or right answers, and that all information would remain anonymous. All responses were requested to be their own. Internet-based devices were not used.

The first mode of data collection was curated questionnaires for each population group. These had three sections that were similar for all participants: (1) demographics and definitions of different age groups, (2) learning outcomes and (3) the visual and audio quality of the video calls and their experience. Each section of the questionnaire had a title page and up to five bullet points about the purpose of questions in that section. Statements at the top of each section provided instructions about the length of answers such as a single sentence, a short paragraph, and a timeline for completion prior to or after each video call.

#### Student data

The three questionnaire sections for students were completed in their classroom. This had three questions requiring them to select an age group (shown in decades from 0–110) for the terms young, old and really, really old. A single sentence with a definition for each of these terms was requested. Additional questions regarded their knowledge of older family members or friends of the family. They were required to write the names of these people along with their age(s) if known. Students were also asked to write words or phrases they used to describe old people. While no data were collected on the level of contact students had previously had with older people, the student questionnaire revealed that all students knew 'old people' in their family (such as grandparents or great-grandparents) and old people who were friends of their family.

In the second section, three questions used the Likert scale. Students were to indicate their level of enjoyment of the video calls, if they had learnt anything and their interest in participating in another video call. Short (one-sentence) answers were requested to explain their rating for each question. During each video call, students wrote lists of words and phrases that were new to them as spoken by the older people. After each video call, these lists were discussed with the teacher in addition to the use of vocalizations and non-verbal communication by the older people. These video calls had an average duration of 40 min with a range of 30 to 48 min. Every student then entered their data into the online version of the questionnaire to measure increases in student vocabulary.

The third section of the questionnaire required students to rate the visual and audio quality of the video calls, their level of enjoyment of the interactions, if they learnt anything and if they would like to participate in another video call. Short (one sentence) answers were requested to explain their rating. A multiple-choice question required students to select one or more of the 15 phrases provided, to indicate changes they will make at their age to live a happy and healthy life when they are really, really old. This was followed by a question to explain their selections. Two final questions required students to state if they thought it would be interesting—or not—to work and care for really, really old people.

#### Teacher data

Teachers also completed these three questionnaire sections. An additional section required information about their teaching experience, six questions about the hearing or sight conditions of students, their expectations of this program and a question about

challenging behaviors and learning difficulties of any students. The names of students were excluded. After the last video call for the year, the teacher of the Term 4 class provided new responses to the same questions about changes in the behavior and learning outcomes of students.

Aged care staff

Aged care staff completed the three questionnaire sections above as well as a section to provide information about their respective fields of expertise. This contained eleven questions about their work history and qualifications. Another section required answers about any changes they observed in the participating residents since they commenced these interactions and to comment on whether this program had met their expectations. For the purpose of time efficiency, the questionnaire sessions for aged care staff (Deputy Facility Manager and Wellness Coordinator) were recorded. The average of these two recordings was 41 min with a range of 28 to 54 min. The Wellness Coordinator provided significantly longer responses due to insights gained through her involvement each week in the preparation of residents for the video calls, being present for most of these interactions and participating in many of the post-video call questionnaire sessions with the residents. Due to the unplanned needs of other residents, the Wellness Coordinator was not able to participate in the full duration of each video call or for the entire period of the questionnaire sessions in Term 3 and 4.

Older participants

The residents also completed the first three questionnaire sections stated above, as well as questions on the amount of weekly communication with family and friends. Data were provided independently by residents with assistance only provided if needed. The next five questions were about their schooldays such as their location, number of schools attended, means of getting to school, favorite subject and favorite childhood memory of school. Four more were about languages spoken at home when they were a child and languages they currently speak. Responses were also required about where they lived as a child, a description of their lifestyle at that time, their funniest childhood memory and favorite game at school or outside of school. The residents were also asked if they still liked to learn, what they would like to learn and to state activities they currently enjoy.

The second point of data collection was the recordings of the video calls involving the students and older people. The average duration of these video calls in Terms 3 and 4 was 30 min (ranging from 26 to 34 min) and 43 min (ranging from 26 to 61 min) respectively. On average, 5 to 6 older people participated in Term 3 with a range from 4 to 9 per session) but this increased to 7 to 8 in Term 4 with a range from 4 to 11. There was an equal representation of male and female residents (2 to 4 per session) in Term 3, but in Term 4 an average of 3 to 6 male residents per session participated compared to 2–5 female residents.

Semi-structured interviews and recordings of these sessions were the third point of data collection. Nine students in Term 4 were interviewed about their evaluation of the program and highlights. Questions were based on the second section of the questionnaire "Learning outcomes and experience of the video calls". The quantity and selection of students per interview plus its duration varied at the discretion of their teachers and student availability. The first three interviews consisted of two students per session. This was followed by a group of three students (two female and one male) then two interviews each with one female and three male students. The average recording time was 19 min with a range from 7 to 31 min. These interviews were recorded at the end of Term 4. Two occurred after the third video call and the remainder three weeks after the fifth and final video call for the year.

Taken together, the video recordings, semi-structured interviews with staff and survey data collected from students and the questionnaires triangulated the data to ensure the validity and accuracy of the evidence collected, as well as to construct a chain of evidence to empirically capture the results of the program. Noting the questionnaires provided a

descriptive overview of the program and greater context to the program than otherwise gained. Moreover, the semi-structured interviews conducted with staff allowed researchers the opportunity to gain valuable insight into the data that were recorded (Yin 2004; Denzin and Lincoln 2005). The interviews were conducted before the first video call and after the last video call. Questionnaire data collected from students were written during and at the end of each video call.

### 4.7. Data Analysis

The data were analyzed using different software including a database, word processing, video editing and speech-to-text transcription programs to identify recurring verbal and non-verbal themes of communication. The observational data included the barriers and facilitators to coordinating an intergenerational learning program via VC, evidence about the direction and flow of learning between both age groups and identifying opportunities for further research. Other data from observations about the active and passive participants included notes about the verbal and non-verbal communication and other interactions during the VC that resulted in a mix of enjoyment, delight, humor, attentiveness, social inclusion, belonging, confusion (at times) and surprise.

The analysis of the content also involved the viewing of the video recordings, which consisted of the dialogue (and transcripts) and observations of non-verbal communication. Interpretations of the data were then validated by participants. This process ensured the final analysis and interpretation were valid, credible, and trustworthy (Creswell and Clark 2007).

## 5. Findings

### 5.1. What Are the Necessary 'Ingredients' for a Successful Virtual Intergenerational Program?

Factors that contribute to a successful virtual intergenerational learning program include reciprocity in interactions through questions and answers for both age groups.

Other factors include humor, and discussions being aligned with the chosen school curriculum topic or cultural event. The data written about the discussion topic by students were being used in their curriculum assessment tasks and then graded by the teacher. In addition, comments from both age groups and staff about looking forward to next week's interaction supported the validation of the program's success. Upon analyzing the data for the necessary ingredients of a successful virtual intergenerational learning program, five key themes were identified. These were activities; participant characteristics; environment; use of equipment; and facilitator interaction.

### 5.2. Activities

The activities in this program were structured in that questions about the school topic for that week were provided to the older adults a few days prior to the program. This enabled them and the Wellness Coordinator (the facilitator at the aged care home) to prepare their answers individually or in small groups. This format seemed to work well for the older adults as they displayed comfort and openness in their answers. For example, an elderly female participant was asked by a student about the price of a bag of lollies when she was a child. Her first reaction was to giggle. As she tried to control her giggling, she repeated the question back to the student in a curious but slightly high-pitched tone, 'how much was a bag of lollies?' Her giggling continued which made a few of the students, staff and other older adults smile and giggle.

This moment and experience of giggling and happiness had a 'contagion effect' (Shanker 2013) on the students, teachers and staff in the care home, because the elderly female participant had not yet answered the question, nor had she added any new information that was actually humorous. However, it was her amusement at the question, the sound of her giggling and facial expressions that triggered the students and adults to react in a similar, happy manner. She eventually provided the students with a direct learning outcome when she answered the question by stating that when she was a child, the price

of a bag of lollies was a ha'penny. However, she then firmly stated to the students that it was a rare and very special occasion to have only one lolly. This elderly female participant explained further that not having money for a bag of lollies was because her family and her friends' families were very poor after 'the war' [World War II].

The retelling of her childhood experience and stating that as a child she never had a bag of lollies for her own pleasure surprised the school children. These pieces of information and insights gained by students would not have been gained in a simple conversation without a pointed question asked. Thus, well-planned and structured activities that provide adequate time for both age groups to prepare for the discussion via video conference within a virtual environment were found to be a critical component of the program.

Another crucial element of the program was the engagement of student participants throughout each video call. In the first school term when this program was facilitated, two students at a time were nominated to ask questions to the older adults. These students were seated in front of the laptop's video camera, with the remaining students sitting further back in the room but still in view of the camera. This enabled the older adults at the aged care home to see on their TV, all the students in the classroom setting. While the remaining students were allowed to ask questions, the majority did not and assumed a passive role. The video calls demonstrated that the remaining students were often bored during these interactions as they doodled or looked away.

In the second term, this was rectified by asking them to take a more active role by instructing them to record: (1) new words, (2) new phrases and (3) vocalizations spoken by the older adults such as giggles, laughter, 'mmm' and 'aha' along with (4) observations of non-verbal communication from the older adults like smiles, hand and facial gestures. Then, at the end of each video call the teacher enquired about which student recorded the highest number of new words, phrases, vocalizations and non-verbal communication observations. This 'gamification' strategy appeared to work better because the majority of students in this second school term, remained engaged during all of the intergenerational learning video calls.

These outcomes were similar to other research findings that highlighted the improvements in vocabulary, language, behavior and communication skills as a result of intergenerational programs (Cartmel et al. 2018; KBC Australia 2018; Polat and Kazak 2015). The program was also found to run better when students asked pointed questions rather than an open question to the group of older adults. When students did not mention the first name of the older participant they wanted a response from, all the older adults remained silent as they were confused and unsure about which one of them should speak. While this problem diminished over the course of the program as the participants became more comfortable with each other, it did highlight the importance of directed communication between participants during the early part of the interaction and the relationship-building phase of any program.

Thus, the core lessons found from this program were that in new virtual intergenerational programs, linking the program to existing school curriculum may improve learning, communication, language skills, behavior and vocabulary. This was evident in the quantitative data gathered about the student learning outcomes. The increase in student vocabulary after five weekly questionnaires in Term 4 indicated they had learned 91 new words and 78 new phrases. These findings are conditional upon the program being well planned and structured with the topic and questions from students being provided to the older adults a few days before the video call.

### 5.3. Participant Characteristics

This program involved students and older adults from a range of cultural backgrounds, as well varying cognitive, behavioral and physical conditions in both students and older adults. None of those inhibitors was found to have a negative impact on the program on a whole. In fact, it could be argued that richer interactions were achieved because of the diversity of the program participants. For example, one of the students reported:

"I enjoyed talking with the older adults. I had to repeat my question because one of them had bad hearing and it was sometimes hard to hear them because they had soft voices. But it was fun to learn about what they did at school like the games they played, what their best subject was and what they ate at school. When I asked about the price of a bag of lollies when she was a child, she told me how much, but it wasn't in cents or dollars. My teacher helped me after [the VC] and said it was a currency called 'imperial' that was used in Australia before 1966. The imperial currency also used words like 'pounds', 'shillings' and 'pence'. So, I had to look up these new words and write about them in my assignment."

In addition, the aged care home Facility Manager reported that this selection of older participants was purposeful in nature, which she felt was successful, as it also improved communication and connection between the older participants outside the program:

"When we discussed the school's invitation of the intergenerational learning activity with our staff and the request to provide participants, we met to decide which of our residents would be best suited to participate. We chose six women and three men ranging in age from 74 to 94. When we met with them and explained the purpose of this activity, we gave each of them a copy of the questions that the school students had provided about their Australian history assignment for the video call. These would be asked by the students during the VC and would form the basis for the dialogue. The residents were given these questions a few days before the VC, to provide them time to prepare their responses. The residents met regularly to discuss and write their answers. The staff also got involved! It was a wonderful and energizing experience for all . . . this activity certainly added a wonderful element of 'excitement' into our village. On the day of the videoconference, it was interesting to observe the nine residents and how especially well-groomed and happy they were to be involved. It was a topic of conversation by residents, their families, and staff for many weeks".

In term three 2019, this program started with around 60 students in the same classroom for the video call with the older adults. This was found to be unsuccessful as the large size of these two combined rooms, large number of students, and the limitations of the speakerphone technology did not work well, nor did it assist the teachers—or principal—as facilitators. After a discussion with the teacher and school principal, the student group reduced to around eight participants, and they met for the video call in a smaller room. For these four video calls, the school principal took the role of facilitator. For the subsequent school term (Term 4 2019), a different Year 6 class participated with 29 students in a 'conventional' size classroom. A new speaker phone system was used that consisted of four wireless units. The combination of four speakerphones and 29 students was found to be more manageable and effective because it worked well to help the students communicate to the older adults and achieve the learning outcomes for that week's topic. Thus, a smaller number of participants (29 or fewer students) in a 'typical classroom setting' is recommended to be used in future programs.

*5.4. Environment*

The nature of virtual interactions means the physical environment for the video conferences needs to be carefully planned. In the aged care home, the older adults sat around a large table with comfortable chairs to interact with the students. Initially, the program was set up so that those older adults who were speaking would be closest to the camera and then they would move to let the next few older adults be closest to the video to improve engagement. However, this was quickly found not to be viable due to the duration of the learning program interaction (60 min) and the length of time it would take older adults to adjust. Thus, ensuring that the room is large enough to comfortable fit all older participants who can be seen and heard on the videoconference is important. To rectify the

problem of not gaining a 'close-up' view of each elder, a motorized camera with controls for left to right panning, up and down movement and 'optical zoom' was used.

Similarly, the same strategy of using the camera built into the laptop was used for the younger participants, and as mentioned previously this strategy was not successful as it did not encourage engagement. Thus, the environment, choice of camera (either internal laptop camera or external USB motorized camera), number and placement of speakerphones, minimization of sun light into the room and noise (from metal trolleys carrying cups, saucers and cutlery) and asking staff to not speak loudly when walking past this 'open room area' are critical considerations when setting up the program. From this study, it is recommended that up to 30 students participate in the classroom and up to 12 older adults (four to six active participants answering questions and engaging in conversations while other older adults are present to enjoy or have slightly less active participation) allows for a bidirectional and meaningful communication strategy that can be implemented and where reciprocal learning for both age groups can be achieved.

*5.5. Use of Equipment*

This study initially engaged the use of readily available equipment (laptop and TV) in both environments. However, for the purpose of this research and to ensure good quality video, a motorized camera with remote control to provide close-up images of the older persons, was provided at no cost to the aged care organization. While this was advantageous, this situation is likely to be an exception rather than the norm in aged care facilities globally. It was noted that more than one speakerphone was needed at the student end to facilitate communication, as it was often stated that the older participants struggled to hear the question from the back of the room. In addition, a similar problem presented itself with the older adults who sat further away from the speakerphone. Therefore, in future studies more than one speakerphone is needed at the school classroom and aged care home. Fortunately, by the time this study was conducted in 2019, freely available consumer grade video conferencing solutions such as Skype and Facetime, as well as high-speed internet on low-cost mobile devices like smart tablets and laptops had been growing in popularity (Teo et al. 2019). This has resulted in an increasing number of aged care home's healthcare staff and residents gaining knowledge in using these technologies for video-based tele-health consultations (KBC Australia 2018; Royal Australian College of General Practitioners 2018). Both commercial-grade and consumer-grade VC solutions also have the facility to invite authorized family members, such as an adult son or daughter, or nominated carers to participate in a video-based tele-health consultation. This enables all participants to ask questions about the health condition of the patient, resulting in more efficient communication. Thus, the availability and useability of the videoconferencing software was not a barrier in this study. Interestingly, while the staff from the nursing home were comfortable using this equipment, the three teachers in the primary school and the older adults were not confident with the technology.

Specialized training was undertaken for teachers prior to implementing this program for all staff involved. This training focused on transitioning from teacher to facilitator in this program. This meant addressing issues such as over prompting, embracing silence, and changing the teacher–student communication dynamic for the duration of the program. Importantly, the goal of the program was to let students develop their own conversational style and give them time to think of a response. The teacher also learnt about the optimal positioning and height of the laptop so the internal (or external) camera could view the two students seated closest to it (within a meter) and the remainder of students in the classroom. This required instruction about seating students in a 'V' shape or cascading rows of chairs to ensure most—if not all—student faces could be seen by the camera and residents.

Training was also provided for the Lifestyle Coordinator, which included learning how to use the video conference application, connection to a TV and basic trouble shooting if there was a problem during a video call such as un-muting a speakerphone or the microphone icon on the video conference application. The Lifestyle coordinator was further

shown how to ensure appropriate seating of elders in the room and adjust the lap top height accordingly.

For students, special attention was placed on ensuring that all student speakers adjusted their sound level when speaking to be well understood. They were also taught about varying their intonation and expressiveness when speaking.

Specific training for the co-facilitators included the location of the speakerphone (a combination microphone and speaker) and instructions about speech during a video call. Topics covered the rate of speech (i.e., to speak slower), consistent volume level, clear pronunciation of syllables and leaving spaces between words and sentences. This training was also provided to the students.

In all cases, training had to occur prior to the program to ensure that (a) the use of equipment was possible at both locations (the school and aged care home) and (b) the respective staff members were comfortable about participating in this program. During the video call, the facilitator at each location tended to begin and end the session, but for the majority of the interaction each facilitator reduced their interaction. This enabled participants from both locations to direct most of the discussion.

### 5.6. Facilitator Interaction

Facilitators were identified as being a teacher, a principal/associate principal of the school and the Wellness Coordinator at the aged care home in this study. In all cases, training had to occur prior to the program to ensure that (a) the use of equipment was possible at both locations (the school and aged care home) and (b) the respective staff members were comfortable about participating in this program. During the video call, the facilitator at each location tended to begin and end the session, but for the majority of the interaction each facilitator reduced their interaction. This enabled participants from both locations direct most of the discussion.

In summary, future virtual intergenerational programs need to consider these five elements carefully in their preplanning to ensure a successful interaction and program is undertaken: school curriculum topic activities; participant characteristics; environment; use of equipment; and facilitator interaction.

### 5.7. What Is the Percieved Impact of Virtually-Run Intergenerational Programs on Participants?

This program was found to have a positive effect on participants. For example, one male student in Year 6, who had been assessed as being on the autism spectrum and with the learning ability of a Year 3 student, was asked to prepare questions for the video conference scheduled for a Thursday morning. The teacher was very reticent to let this student have such a responsibility, due to his disruptive and unpredictable behavior. In addition, the teacher was concerned about potentially upsetting the older adults if this student became enraged. To the pleasant surprise of the teacher—and other Year 6 students—this 11-year-old male student actually engaged in an interesting conversation for nearly four minutes with a 95-year-old male participant, who we will refer to as 'George'. After the videoconference and in the presence of the teacher, the primary author interviewed the student. The student was asked to describe any highlights he had from the video conference. He stated that he enjoyed speaking with George because he had fond memories of speaking with his grandfather and that he missed these conversations due to the passing away of his grandfather two years ago. Upon enquiring if the student had other questions for George, the student replied he would like to know what age his mother and father died. The student's response to the third and final question about any other highlights from the video conference nearly caused the teacher to fall off her chair! He stated that he "could see that George had a real passion for education". His insight and use of such words astounded the teacher, more so his behavior in the classroom after the video call improved significantly with no disruptive outbursts, which was atypical for this student.

Similarly, another student who was usually well behaved but shy in class, refused to volunteer for an activity in class. However, over the course of the program this changed

to the point where he volunteered to host the last video call with the older adults. This spontaneity and motivation to participation was out of character for him. The teacher reflected on how proud she was to see such growth and self-confidence develop in this student over the course of the program. Across all students the teacher identified a number of improvements in their behavior and self-confidence, which she considered to be an outcome of the program.

The facility manager of the nursing home also reported the program to be a "calming and socializing intervention" for all participants. She provided several examples about a significant change she had observed in older adults and staff throughout the program in her interview. One example is a 93-year-old male participant. He would meet with the Facility Manager 'every other day' to complain about her management, the meals and chef, even though the meals were excellent, and she was a good manager, as evidenced by positive staff, family and other client feedback. His comments at mealtimes would result in a 'near riot', resulting in this older adult not eating his meal and upsetting other older adults who would often lose their appetite. Consequently, the 93-year-old male began losing weight. He was often—if not always—grumpy, confrontational, and negative towards staff, other older adults and his family. However, after two weeks of video calls, the observations by the Facility Manager and Doctor of this male elder included an end to his complaints, a happier disposition, socialization with staff and other older adults and an increase in weight. The Facility Manager also observed of a female older adult, a reduction in nighttime nurse call requests which she equated to better quality sleep. Overall, the older adults involved in the intergenerational-learning video calls were reported to be calmer, happier, socialize more with each other and staff, appear less depressed and they 'lingered' after meal times in conversation rather than immediately returning to their rooms. These observations about improvements in behavior and communication concur with the research on co-located (or shred site) intergenerational learning programs. These interactions highlight the value of intergenerational programs in terms of improving social interactions, behaviors, vocabulary, and engagement for all participants during the study.

*5.8. Can Both Generations Use, and Benefit from, Virtual Intergenerational Programs to Improve Social Inclusion and Reduce Social Isolation?*

This study focused on how the program improves social inclusion and reduces social isolation. The video and interview data were powerful mechanisms to review this question. Prior to the intergenerational learning video calls, all residents involved in these interactions voluntarily socially isolated after mealtimes and activities. Since their participation in this program, staff have observed these residents talking more often with each other. For example, the dining room would normally be vacated before staff cleaned these areas. Since the video calls, the participating residents choose to remain in the dining room and engage in conversation after their meals, whereas prior to this program they would return immediately to their room. Thus, the findings revealed that the intergenerational programs across both terms served as powerful reminders of the past, as well as highlighting the importance of connection between generations by engaging participants of all ages in new experiences. As societies grow, it is a regular occurrence that we disengage in connecting with vulnerable populations. This study found that virtual intergenerational programs can, and do, have a positive influence on social inclusion in order to reduce social isolation.

## 6. Summary

Intergenerational programs provide a powerful and meaningful way to connect generations in a purposeful manner. The findings of this 'virtualized' form of intergenerational learning created a 'virtualized' village. This study provided a unique insight into the impact that a virtual intergenerational program has on social isolation, inclusion, and participant outcomes in an Australian setting. The results of this program highlight that including those with behavior and cognitive challenges, as well as those from a diverse background, result in the same, if not more benefits than not including them. However, it

should be noted that, as with all qualitative research, several limitations exist. For example, the results of this research are not generalizable for application to a larger section of the population, because it cannot be exactly replicated. This includes the small amount of quantitative data obtained in this study, as it was not from random sampling and was designed to be descriptive only in its support of the qualitative results. This is due to the changing dynamics that occur between each of the participants and groups during the phenomenon of intergenerational learning and communication (Tracy 2012). However, this qualitative research has 'transferability', meaning that the components of the research can be transferred to another group of participants at another place, time, and context (Tracy 2012) and as such, we believe the results of this study are important to communicate widely. Notwithstanding current publications and knowledge, this study also highlights the possible benefits of intergenerational programs on the disabled, which has received limited attention in the intergenerational literature to date.

A second limitation is that, for qualitative research, a small sample size is often an issue because sampling and the size of sampling only provides a representation of the participants and objects that would affect the outcomes of the research (Baxter and Jack 2008). However, for the purpose of this research, a sample size of 29 students was determined by the school principal, classroom teacher and older staff to be manageable and feasible. The sample size of nine older adults was determined by the Facility Manager at the residential aged care home to also be a manageable and feasible number for this project and for similar projects in the future. It is pleasing to note that at times there would be more than nine older adults in the video call. Some would not participate, one was blind, another older adult was hard of hearing, but they all enjoyed hearing or seeing the children on the TV screen or just being part of the group experience. Thus, while caution should be taken when interpreting these results to a wider population group, we believe that the sample from this study is appropriate to publish and communicate broadly (Tracy 2012).

Finally, data were collected and administrated by the lead author of this study who also provided facilitation within the program. It is noted that potential researcher influence may have occurred, and this research should be replicated to further explore the transferability of the findings beyond the scope of this study. Thus, care should be taken when interpreting the results.

In conclusion, this study presents a new approach to intergenerational learning. It argues that the use of video conferencing engaged school students in a typical classroom and nonfamilial older adults living at a residential aged care home, in purposeful research and discussion on a mutually agreed topic that could achieve pre-defined learning outcomes for both age groups. In addition, the effect of face-to-face communication using video conferencing created an innovative environment and new framework for the sharing of knowledge as a reciprocal educational, enjoyable, and at times humorous, interaction between these two age groups. More importantly, the study highlighted the continued value of intergenerational social interactions that are even more crucial in a global pandemic such as COVID-19, through virtual means.

**Author Contributions:** Conceptualization, G.C., J.A.F., K.R. and G.D.P.; methodology, G.C., J.A.F., K.R. and G.D.P.; software, G.C., J.A.F., K.R. and G.D.P.; validation, G.C., J.A.F., K.R. and G.D.P.; formal analysis, G.C., J.A.F., K.R. and G.D.P.; investigation, G.C., J.A.F., K.R. and G.D.P.; resources, G.C., J.A.F., K.R. and G.D.P.; data curation, G.C., J.A.F., K.R. and G.D.P.; writing—original draft preparation, G.C., J.A.F., K.R. and G.D.P.; writing—review and editing, G.C., J.A.F., K.R. and G.D.P.; visualization, G.C., J.A.F., K.R. and G.D.P.; supervision, G.C., J.A.F., K.R. and G.D.P.; project administration, G.C., J.A.F., K.R. and G.D.P.; funding acquisition, G.C., J.A.F., K.R. and G.D.P. All authors have read and agreed to the published version of the manuscript.

**Funding:** This research did not receive any external funding.

**Institutional Review Board Statement:** The Office for Research at Griffith University provided ethical clearance and full approval for this research project originally titled "Intergenerational Learning using video conferencing" The approval number is 2019/683.

**Informed Consent Statement:** Informed consent was obtained from all subjects involved in the study.

**Data Availability Statement:** Not applicable.

**Conflicts of Interest:** The authors declare no conflict of interest.

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
