# Peer review of "The Impact of a Virtual Environment for Intergenerational Learning"

_socsci, doi:10.3390/socsci12030147_

Round 1

Reviewer 1 Report

See file word

Reviewer 2 Report

Overall, this study is timely and impactful. It provides a needed focus on an underserved population. It also pertains to an ongoing need, that is especially crucial now, related to social inclusion / social isolation. The detailed descriptions of participant comments are informative and much appreciated, as are the descriptions of the intervention and how it evolved / improved over time. Thank you for your important contributions!

Abstract

-Add clarity on how the findings were obtained (interviews)

-Includes variables / DVs that were not covered in the rest of the manuscript and that are missing from the opening of the Intro. section

---For example, “overall health” and “wellbeing” were mentioned here and then not defined, measured, or reported on

Introduction

-Stated in several locations that the younger participants ranged from 11-12 years of age, but also said “…students aged 5 to 18 located in their classroom” – fix that if it is a typo, or clarify

-Overall, the Introduction section provides an overview of related research findings, but there is a need to clarify who the authors are referring to when they mention “older adults” – it would be helpful to state when the older adults in the cited studies were active community-dwelling older adults, frail residentially-dwelling older adults, or otherwise

Methods / The intergenerational program components and process

-Each of the groups behind the two main students also interacted with the older adults during each video call by asking questions or responding to comments from the older adults.

            ---Can the authors provide data on the frequency of that occurring / which students participated how often?

-At the end of video call each week, a questionnaire was completed in writing by the students. Questions ranged from the quality of the video, sound, and speech to highlights of the video call and suggestions to improve the next interaction.

            ---Clarify if /how the data from this “questionnaire” pertains to the following comments (overall issue)

Methods / Participant characteristics

-Add details about the age of the children – mean age, standard deviation, and range

-Can the authors provide data on the students level of / amount of contact with older adults? their grandparents?

-Add details about recruitment / clarify whether or not all students in those classes and/or all older adults at that location were encouraged to participants, and then the actual sample just includes those who consented – any way to address potential differences between those who chose to participate and those who did not?

-Any inclusion criteria for the older adults? – said “good cognitive and speech skills” – determined that how?

-Issues related to the creation of the sample of older adults is covered in the findings section, but not here, when is seems to belong in this section instead

---Address potential issues related to the generalizability of the current findings if the samples were not created with random selection / random assignment, but created by those staffing the intervention and/or according to who wanted to participate in an intergenerational program

Methods / Data Collection

-This section is missing a needed detailed description of the questions and/or statements in the measures (surveys, interviews, questionnaires, etc.)

-Clarify if any quantitative data was collected / could be calculated

-Said  “Survey data collected from students was written…” – written by the students or just the researchers?

-Did the participants provide any direct data / respond to any questions anonymously by themselves?

---Need to address potential issues with accuracy / quality of data if / when participants could have been influenced by the researchers / presence of the researchers

Methods / Data analysis

-Provide details about why said the analysis  interpretation was “…likely to be transferable to other contexts

-Provide clarity for each source of data – data from surveys, data from interviews, and data from any other sources, as all that is mentioned here is described as observational data

---Again, there is a need to address potential issues with the accuracy / quality of the data if everything was interpreted through the lens of the researchers and no anonymous direct participant data was collected by administering measures to the participants themselves that could then be deidentified

FINDINGS / What are the necessary ingredients…

-The authors cannot provide evidence that “…rare insights” occurred --- seem to be overstating the findings

---This also relates to the potential influence of existing level / amounts of contact with older adults / grandparents mentioned above

-The authors again appear to be overstating the findings with their statements:

“…could measure increases in the vocabulary, language and communication skills of these students…”

“…improves learning, communication, language skills, behaviour and vocabulary

            ---No established measures of those variables were used in the current study

Can both generations use, and benefit from, virtual intergenerational programs…

-The content in this section needs to be located instead in an overall Discussion section

---The inclusion of that Discussion section would provide clarity / distinctions between the actual findings / results from the current study and the authors’ subsequent interpretations of those findings, along with a broader discussion of the potential implications of those findings / interpretations

Strengths and Limitations

-Please add specific issues noted above that are not already addressed

-If questions / concerns included above cannot be settled, then please add them as potential limitations in the discussion section

Author Response

Please see the attchment

Reviewer 3 Report

Congratulations for the choice of topic and scientific quality of the article.

Reviewer 4 Report

Dear authors,

Thank you for letting you read about your interesting study and project of intergenerational virtual encounters. I believe it is a project worth publishing on. I have a few comments to strengthen the manuscript.

1. Take care to situate and discuss qualitative research in an appropriate manner. Reliability and validity are differently assessed than in quantitative research and should be discussed according to standards for qualitative research both in the methods and in the limitations. This becomes particularly clear in the limitations (sample size is not a consideration by itself, but rather the inclusion of diverse perspectives on the issue at hand), but also in other parts of the article. E.g. qualitative research is not meant to answer questions of quantitative nature such as how many, effect, impact sizes etc. It can answer how and why questions which to me are more interesting anyway. In that regard, please consider rephrasing: effective (that cannot be measured with qualitative research), impact (or add an adjective, such as perceived impact), succesful (what is succesful?), positive effect etc.

2. The authors state that they examine "how we can address social isolation through a virtual intergenerational programme" but I find the manuscript about a lot, but not that. Social isolation is not measured, nor asked about in the interviews with students and/or older adults. Or at least that is not clear from the manuscript. Thus a statement such as 'positive influence on social isolation' is not supported by the provided data/findings in this manuscript and I find little trace in the methods that clarify if social isolation has been mapped/measures/asked about before and after this intervention nor is it clear from statements and/or data presented of the older adults and/or the students.

In fact it seems much more about 'what works/is needed for a virtual intergenerational programme to work' and that is interesting by itself. So perhaps leave the social isolation behind.

3. On methods: I was struck by the fact that the older adults themselves were not asked about their experiences and I find this a major limitation that needs mentioning. I was also missing the topic list for the semi-structured interviews and the survey questionnaire for the students. I also missed the average length of the interviews, the way the data was handled (transcribed?), analysed (i.e. grounded theory? thematic? interpretative? content analyses? and how coding was done? several people? one person? code book? how were the videos analysed? which software was used? how was the interpretation of the data then presented to the participants? And for the findings: what additional insights were gained from that? who did data collection, interviews and who was involved in data analyses? I am also a little uncertain about the 'immersive fieldwork' where was the researcher(s) during the videocalls? what was 'immersive' about it? how was the data collected?), in brief I need much more information to assess the quality of the qualitative research in the methods section.

4. On the findings: I find the use of factors (page 6, lines 288 and further) and the elements somewhat confusing. I also find that the headings of the elements are somewhat unclear, since they have no classification. As the guiding question posed by the authors is: what are key components for successful intergenerational virtual encounters? then not activities, but well-planned and structured activities. Not participant characteristics but what about participant characteristics etc. (and perhaps gamification deserves a seperate §). 

4. I am missing a dicussion that situates the findings in the literature and discusses critically how it fits with earlier studies and findings (e.g. mentioned in the introduction) and how this study diverts from that or adds to our current knowledge.

less important suggestions:

5. Perhaps more can be said about the training mentioned on page 10.

6. the abstract overstates what the study supports (and there is no need for that) "The findings reveal that, to improve overall health, socialisation and wellbeing of all participants"...

7. In the introduction, perhaps it seems more logical here to mention the ageism campaign instead of the healthy ageing agenda of the WHO (or perhaps both).

8. Consider some more reflection on the inclusion of younger individuals in this programme, it seems to work really well with students that are prepared, are able to reflect and integrate these experiences but may be less suited for younger or older students? perhaps the authors could say something more on this in the discussion or limitations? Also in regard to having a rather 'innovative' aged care setting with rather good equipment.
